# Diagnostic value of long noncoding RNAs as biomarkers for Ankylosing Spondylitis: A systematic review and meta-analysis

Ermiyas Alemayehu[1]*, Sintayehu Ambachew[2,3], Daniel Asmelash[4], Melaku Ashagrie Belete[1]

**1** Department of Medical Laboratory Sciences, College of Medicine and Health Sciences, Wollo University, Dessie, Ethiopia, **2** Department of Clinical Chemistry, School of Biomedical and Laboratory Sciences, College of Medicine and Health Sciences, University of Gondar, Gondar, Ethiopia, **3** Adelaide Medical School, Faculty of Health and Medical Sciences, The University of Adelaide, Adelaide, Australia, **4** Department of Medical Laboratory Science, College of Medicine and Health Sciences, Mizan-Tepi University, Mizan-Aman, Ethiopia

* ermiyas0009@gmail.com

## Abstract

### Introduction

Ankylosing spondylitis (AS) presents as a debilitating form of arthritis with potential for severe damage. In chronic cases, patients may experience progression to ankylosis and spinal immobility, significantly diminishing their quality of life. Given these challenges, there is a pressing need to explore novel diagnostic targets. Thus, this study aimed to evaluate the diagnostic potential of long non-coding RNAs (lncRNAs) for AS.

### Methods

The search encompassed various databases including PubMed, Scopus, Embase, and Hinari. Analysis of pooled sensitivity, specificity, positive likelihood ratio (PLR), negative likelihood ratio (NLR), diagnostic odds ratio (DOR), and area under the curve (AUC) was carried out using Stata 17.0 software, employing a random effects model. Heterogeneity among studies was assessed through the Cochran-Q test and $I^2$ statistic tests, followed by subgroup analyses to delve into primary sources of heterogeneity. Publication bias was assessed using Deeks' funnel plot, while Fagan's nomogram was used to evaluate the clinical utility of lncRNAs for AS. Furthermore, a sensitivity analysis was conducted to assess the robustness and reliability of the findings.

### Results

This systematic review and meta-analysis synthesized data from 11 articles, of which 6 were included in the meta-analysis. The pooled sensitivity, specificity, PLR, NLR, DOR, and AUC of lncRNAs for diagnosing AS were calculated as 0.81 (95%

**Data availability statement:** All relevant data are within the manuscript and its Supporting Information files.

**Funding:** The author(s) received no specific funding for this work.

**Competing interests:** The authors have declared that no competing interests exist.

CI, 0.73–0.88), 0.81 (95% CI, 0.55–0.93), 4.2 (95% CI, 1.64–10.77), 0.23 (95% CI, 0.17–0.32), 18.1 (95% CI, 6.39–51.24), and 0.86 (95% CI, 0.83–0.89), respectively. Furthermore, subgroup analysis revealed that lncRNAs identified in peripheral blood mononuclear cells (PMBCs), those showing upregulation, studies utilizing β-actin as the internal reference control, and research involving AS patients from China demonstrated enhanced diagnostic accuracy for AS.

## Conclusions

In conclusion, the existing evidence indicates that lncRNAs have substantial diagnostic value in predicting AS and can serve as effective non-invasive markers for the condition. However, the results should undergo further validation through well-designed longitudinal studies with larger sample sizes in the future to enhance their reliability and generalizability.

## Introduction

Ankylosing spondylitis (AS) is a form of spinal arthritis primarily impacting the spine, sacroiliac joints, spinal attachment points, and other axial bones. It leads to persistent inflammatory damage and joint function deterioration [1]. It affects approximately 0.09%−0.3% of the global population and predominantly manifests in young individuals [2,3].

Currently, AS lacks a definitive cure, and pharmacological therapy stands as the primary approach, particularly for severe cases where surgical intervention may be necessary [4,5]. Diagnosing AS is challenging due to symptom overlap with other conditions, the limitations of imaging techniques like X-rays and MRIs [6], and the late appearance of hallmark radiographic changes [4,7]. While biomarkers such as HLA-B27, CRP, and ESR are used, they lack sensitivity in many cases [8], highlighting the urgent need for more reliable diagnostic tools to enable early detection.

Long non-coding RNAs (lncRNAs), exceeding 200 nucleotides in length and lacking protein-coding capacity, exert post-transcriptional influence on gene expression [9]. Studies highlight lncRNAs' significant regulatory roles in diverse biological processes, including cell division, apoptosis, and the release of pro-inflammatory cytokines [10–12]. Examining the expression patterns and roles of lncRNAs could aid in disease treatment and prediction [13,14]. Recently, mounting evidence suggests that lncRNAs actively contribute to the development of orthopedic diseases, including scoliosis, intervertebral disc herniation, arthritis, and AS [15–17]. LncRNAs have gained attention as promising biomarkers for various diseases, including AS, due to their tissue-specific expression, regulatory roles in immune and inflammatory pathways, and stability in body fluids.

They possess the capacity to enhance the reliability, sensitivity, and specificity of molecular techniques in clinical diagnosis [18]. While substantial evidence suggests that lncRNAs might play significant roles in the pathogenesis of AS by modulating inflammatory and immune pathways, and offer a promising diagnostic biomarker for

AS [19–21], previous research has not definitively established their clinical diagnostic significance in AS due to disparities in study design, expression levels, specimen type, and sample size. Therefore, this systematic meta-analysis aimed to evaluate the diagnostic utility of long noncoding RNAs (lncRNAs) as biomarkers for AS by systematically reviewing and analyzing existing literature.

## Methods

### Study protocol registration

This systematic review and meta-analysis adhered to the guidelines provided in the Preferred Reporting Items for Systematic Reviews and Meta-Analyses (PRISMA) Statement [22] (**Table** in S1 Checklist). Furthermore, the study protocol has been formally registered with the International Prospective Register of Systematic Reviews (PROSPERO) under the registration number CRD42024495975.

### Search strategy

We implemented a comprehensive and systematic search strategy to identify potentially relevant articles across multiple electronic databases, including PubMed, Scopus, Embase, and Hinari, up to March 15, 2025. By employing Medical Subject Headings (MeSH) terms and Boolean operators ("AND" and "OR"), we utilized a range of keywords, such as "long noncoding RNA*" OR "long non-coding RNA*" OR "long non coding RNA*" OR "lncRNA*" OR "linRNA*" AND "biomarkers" OR "diagnostic value" AND "Ankylosing Spondylitis" OR "Ankylosing Spondyloarthritis" OR "Ankylosing Spondylarthritides" OR "Bechterew Disease". Moreover, to ensure the comprehensiveness of our search, we meticulously scrutinized the reference lists of relevant articles for any additional publications that might have been overlooked in the initial database search. The detailed search methodology, including search terms and strategies, is presented in **Table** in S1 Table.

### Selection of included studies

After completing the literature search, we imported articles identified through the systematic search into Endnote 20 software for screening of eligible studies. Once duplicates were removed, two authors (EA and MAB) independently screened the titles and abstracts of the studies to identify all eligible articles. Following this, we obtained the full texts of the included studies and independently reviewed them according to the predefined inclusion and exclusion criteria. Any discrepancies in the data were assessed by a third researcher (SA), and any disagreements were resolved through discussion.

### Inclusion and exclusion criteria

In accordance with the established inclusion and exclusion criteria, eligible articles were included based on the following requirements: First, the studies had to focus on lncRNAs for diagnosing AS patients. Second, complete data were required in each study to calculate the values of true positives (TP), false positives (FP), false negatives (FN), and true negatives (TN). Third, the studies needed to be original research conducted on human subjects and published in peer-reviewed journals. Lastly, the studies considered for inclusion were observational, including cross-sectional, cohort, and case-control studies. Conversely, studies were excluded if they were reviews, not peer-reviewed, letters, commentaries, case reports, or case series, or if they lacked sufficient information to calculate the required diagnostic parameters.

### Data extraction

Two reviewers (EA and DA) independently conducted data extraction from the included studies using a Microsoft Excel sheet. The extracted data encompassed various details, including the first author, year of publication, country, sample size of participants (AS patients and the healthy individuals), specimen type, diagnostic methods, internal reference control,

lncRNA type, expression of lncRNAs, cut-off values, sensitivity, specificity, and area under curve values (AUC) (**Table in** S2 Table). In cases of disagreements, an independent third researcher (MAB) assumed responsibility for resolving disputes.

### Quality assessment

Two authors (EA and MAB) independently utilized the Quality Assessment of Diagnostic Accuracy Studies-2 (QUADAS-2) tool [23] to assess the quality of the 11 included articles. This tool evaluates four domains: patient selection, index test, reference standard, and flow and timing, assigning each domain a rating of high (red), unclear (yellow), or low (green) risk bias using Review Manager (RevMan) 5.3 software. Similarly, concerns about applicability within the first three domains were also evaluated using the same rating scale of high, unclear, or low risk bias. Any discrepancies were resolved through discussions involving a third author (SA).

### Data analysis

We conducted meta-analyses using STATA 17.0 software to evaluate the diagnostic potential of lncRNAs for AS. Our analysis included pooled estimates of sensitivity, specificity, negative likelihood ratio (NLR), positive likelihood ratio (PLR), diagnostic odds ratio (DOR), and the area under the summary receiver operating characteristic curves (AUC of the SROC), each with corresponding 95% confidence intervals (CIs). Additionally, we applied the Hierarchical Summary Receiver Operating Characteristic (HSROC) model to assess the overall diagnostic performance across studies. Heterogeneity was examined using the Cochrane Q test and $I^2$ statistics [24], with $I^2 > 50\%$ and $p < 0.05$ indicating significant heterogeneity, warranting the use of a random effects model. We also investigated threshold effect-induced heterogeneity using the ROC plane and Spearman correlation coefficient. Subgroup analyses were undertaken to identify potential sources of heterogeneity, while sensitivity analysis was conducted to assess the stability of our results. Moreover, we employed Deek's funnel plot asymmetry test to explore the presence of publication bias, considering a significance level of $P < 0.10$ as suggestive of bias [25]. Additionally, Fagan's nomogram was utilized to evaluate the clinical utility of lncRNAs as a diagnostic tool.

## Results

### Study selection process

An initial literature search identified 311 articles across four databases: PubMed, Scopus, Embase, and Hinari, as well as through a direct Google search. Out of these, 66 articles were eliminated due to duplication, resulting in 245 articles for screening of titles and abstracts. Following the review of titles and abstracts, 221 articles were deemed irrelevant and excluded, leaving 24 articles for full-text assessment. After the full-text assessment, thirteen articles were excluded for various reasons, leaving 11 articles. All 11 articles met the criteria for qualitative analysis (**Table in** S3 Table). Ultimately, six articles were included in the quantitative analysis (meta-analysis). The screening process is illustrated in **Fig 1**.

### Characteristics of included studies and quality assessment

This study involved 11 articles [19–21,26–33], out of which 6 [21,29–33] were used for the meta-analysis, encompassing a total of 16 studies (8 for the meta-analysis). It included 808 patients diagnosed with AS and 577 healthy controls (375 AS patients and 288 healthy individuals for the meta-analysis). Among these articles, nine originated from China [19–21,26–31], while the remaining two were from Egypt [32,33]. The specimens varied: four articles utilized polymorphonuclear blood cells (PMBCs), one employed serum samples, two utilized both serum and open sacroiliac biopsies, and two utilized plasma samples. Across all studies, quantitative real-time polymerase chain reaction (qRT-PCR) was uniformly employed

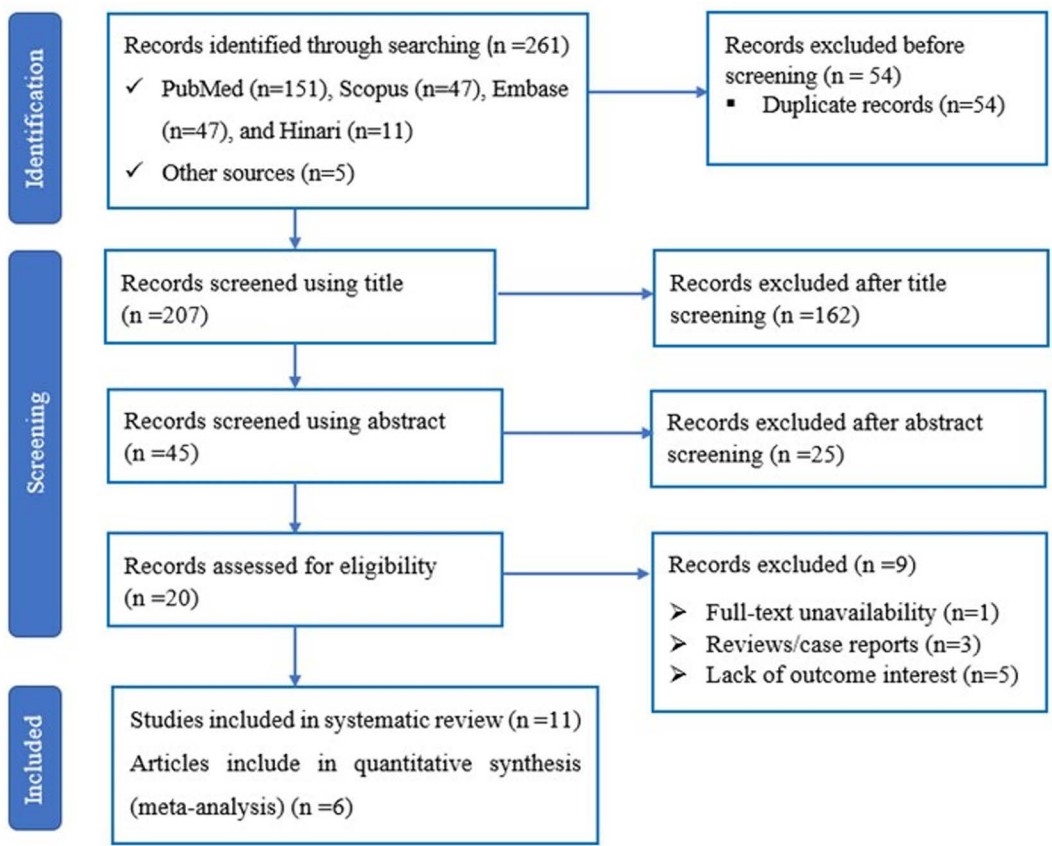

**Fig 1. Flow diagram summarizing the selection of eligible studies.**

to detect levels of lncRNAs. Regarding internal reference control, seven articles used β-actin, one used U6, one used GAPDH, and one used 18S rRNA. Across these screened articles, a total of 12 lncRNAs exhibited differential expression between AS patients and healthy controls. Among them, 9 lncRNAs (LINC00311, H19, LOC101929023, 326C3.7, Lnc-ITSN1–2, Linc00304, Linc00926, MIAT, and NONHSAT227927.1) were up-regulated in AS patients compared to controls, while 2 lncRNAs (AK001085 and MEG3) were down-regulated. Notably, lncRNA TUG1 demonstrated both up and down-regulation in AS patients. **Table 1** outlines the characteristics of all the studies.

The assessment of the quality of the articles included in the study was conducted using a modified version of the QUADAS-2 tool with RevMan 5.4. **Fig 2** displays the graph depicting the risk of bias and applicability concerns for the included articles.

## Diagnostic value of LncRNAs for AS

A random-effect model was chosen for evaluating the diagnostic accuracy of lncRNAs in AS patients due to the $I^2$ value exceeding 50% (70.47% for sensitivity and 92.69% for specificity). The combined outcomes were as follows: sensitivity, 0.81 (95% CI, 0.73–0.88); specificity, 0.81 (95% CI, 0.55–0.93) (**Fig 3**); PLR, 4.2 (95% CI, 1.64–10.77); NLR, 0.23 (95% CI, 0.17–0.32) (**Fig 4**); DOR, 18.1 (95% CI, 6.39–51.24) (**Fig 5**); and AUC from SROC curve, 0.86 (95% CI, 0.83–0.89) (**Fig 6A**). The results obtained from the HSROC model (**Fig 6B**) revealed a β estimation of 1.15 (95% CI = 0.35–1.94) and z = 2.83, p = 0.04. The λ estimation stood at 3.41 (95% CI, 2.53–4.29). These findings suggest that lncRNAs hold promise

**Table 1. Baseline characteristics of the included studies in this systematic review and meta-analysis.**

| Authors | Year | County | LncRNAs | Expression | Specimen | Method | Reference | Participants | | Cut-off | Sen (%) | Spe (%) | AUC |
|---------|------|--------|---------|------------|----------|--------|-----------|------|------|---------|---------|---------|-----|
| | | | | | | | | AS | HC | | | | |
| Li X et al [31] | 2017 | China | AK001085 | Downregulated | Serum | qRT-PCR | β-actin | 117 | 76 | 0.186 | 62.9 | 93.6 | 0.868 |
| Lan et al [20] | 2018 | China | TUG1 | Downregulated | Serum | qRT-PCR | β-actin | 82 | 32 | NA | NA | NA | 0.796 |
| Lan et al [20] | 2018 | China | TUG1 | Downregulated | Biopsies | qRT-PCR | β-actin | 34 | 32 | NA | NA | NA | 0.891 |
| Zhong et al [19] | 2019 | China | LINC00311 | Upregulated | Plasma | qRT-PCR | 18S rRNA | 80 | 80 | NA | NA | NA | 0.904 |
| Liu et al [26] | 2019 | China | MEG3 | Downregulated | Serum | qRT-PCR | β-actin | 172 | 98 | NA | NA | NA | 0.748 |
| Liu et al [26] | 2019 | China | MEG3 | Downregulated | Biopsies | qRT-PCR | β-actin | 42 | 36 | NA | NA | NA | 0.886 |
| Zhang et al [28] | 2020 | China | H19 | Upregulated | PBMCs | qRT-PCR | β-actin | 49 | 49 | NA | NA | NA | 0.653 |
| Zhang et al [28] | 2020 | China | LOC101929023 | Upregulated | PBMCs | qRT-PCR | β-actin | 49 | 49 | NA | NA | NA | 0.637 |
| Wang et al [29] | 2021 | China | 326C3.7 | Upregulated | PBMCs | qRT-PCR | β-actin | 68 | 29 | NA | 80 | 60.5 | 0.739 |
| Li M et al [30] | 2021 | China | Lnc-ITSN1–2 | Upregulated | PBMCs | qRT-PCR | GAPDH | 63 | 60 | NA | 69.8 | 96.7 | 0.900 |
| Wang et al [21] | 2022 | China | Linc00304 | Upregulated | PBMCs | qRT-PCR | β-actin | 24 | 20 | 0.413 | 90 | 42.3 | 0.687 |
| Wang et al [21] | 2022 | China | Linc00926 | Upregulated | PBMCs | qRT-PCR | β-actin | 24 | 20 | 0.299 | 88.2 | 41.7 | 0.664 |
| Wang et al [21] | 2022 | China | MIAT | Upregulated | PBMCs | qRT-PCR | β-actin | 24 | 20 | 0.432 | 89 | 44.3 | 0.623 |
| Ding et al [27] | 2023 | China | NONHSAT227927.1 | Upregulated | PBMCs | qRT-PCR | β-actin | 50 | 30 | NA | NA | NA | 0.846 |
| Tawfeek et al [32] | 2023 | Egypt | TUG1 | Upregulated | Plasma | qRT-PCR | U6 | 50 | 50 | 6.2 | 88 | 84 | 0.874 |
| Esawy et al [33] | 2023 | Egypt | H19 | Upregulated | Plasma | qRT-PCR | β-actin | 53 | 53 | 1.41 | 81.1 | 100 | 0.955 |

Note: AS: ankylosing spondylitis; AUC: area under curve; GAPDH: glyceraldehyde-3-phosphate dehydrogenase; HC: healthy control; NA: not available; PBMCs: peripheral blood mononuclear cells; qRT-PCR: quantitative real-time polymerase chain reaction.

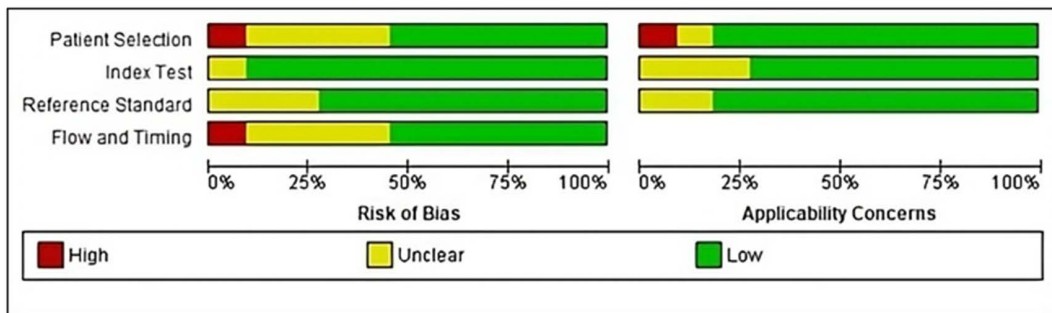

**Fig 2. Quality assessment of eligible studies using QUADAS-2.**

as potential biomarkers for distinguishing AS patients from healthy individuals, with statistically significant results from the HSROC model supporting their diagnostic utility.

## Threshold effect

An assessment was conducted to determine the contribution of the threshold effect in the observed heterogeneity, using the Spearman correlation coefficient and ROC plane analysis. A strong negative correlation between sensitivity and specificity ($p < 0.05$) and a shoulder-arm appearance on the ROC plane typically indicate the presence of a threshold effect. In our study, the corresponding Spearman correlation coefficient was −0.314 with a p-value of 0.09 ($p > 0.05$), suggesting no statistically significant correlation. Additionally, the ROC plane lacked the characteristic shoulder-arm distribution (**Fig 7**). These findings suggest that heterogeneity due to the threshold effect is unlikely, further supporting the reliability of our results.

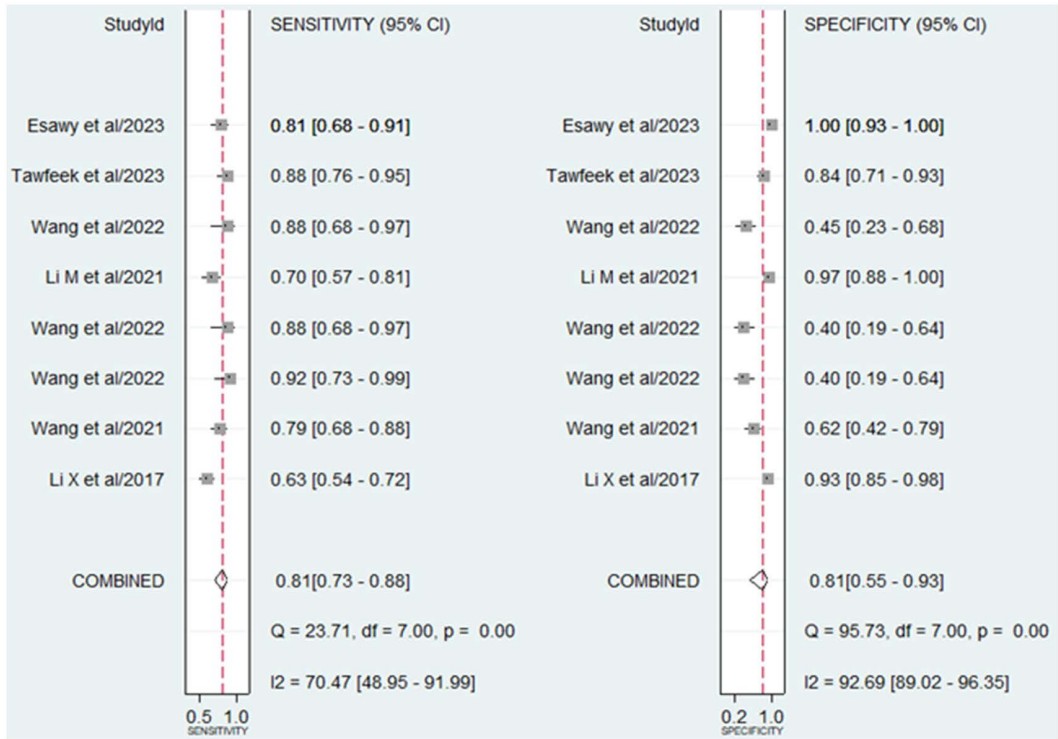

**Fig 3. Forest plots showing the sensitivity and specificity of lncRNAs in the diagnosis of AS.**

## Clinical applicability

Furthermore, a Fagan plot was utilized to evaluate the clinical utility of lncRNAs in diagnosing AS. Initially, the pre-test probability was established at 20%. Subsequently, a positive test yielded a post-test probability of 51% with a PLR of 4.2. Conversely, the NLR was 0.23, leading to a reduction in the post-test probability to 5% following a negative test (**Fig 8**).

## Subgroup analyses

Subgroup analyses were conducted based on country, specimen type, internal reference control, and lncRNA expression mode, as detailed in **Table 2**. Among studies conducted in Chinese populations, lncRNA demonstrated a sensitivity of 0.80 (95% CI, 0.69−0.88), specificity of 0.71 (95% CI, 0.42−0.89), PLR of 2.7 (95% CI, 1.3−5.8), NLR of 0.28 (95% CI, 0.21−0.38), DOR of 10 (95% CI, 4−24), and AUC of 0.83 (95% CI, 0.79−0.86). Subgroup analyses by specimen type showed that studies utilizing PMBCs reported a sensitivity of 0.83 (95% CI, 0.74−0.89), specificity of 0.63 (95% CI, 0.33−0.85), PLR of 2.2 (95% CI, 1.1−4.5), NLR of 0.27 (95% CI, 0.18−0.40), DOR of 8 (95% CI, 3−22), and AUC of 0.84 (95% CI, 0.80−0.87). Furthermore, studies using β-actin as the internal reference control yielded a sensitivity of 0.82 (95% CI, 0.72−0.89), specificity of 0.74 (95% CI, 0.40−0.92), PLR of 3.1 (95% CI, 1.1−8.7), NLR of 0.25 (95% CI, 0.16−0.38), DOR of 13 (95% CI, 4−44), and AUC of 0.85 (95% CI, 0.81−0.88).

## Sensitivity analysis

Additionally, a goodness-of-fit analysis and bivariate normality analysis were conducted, indicating the robustness of our model. Outliers were detected via sensitivity analysis, and it was found that no outliers were present, suggesting that the pooled results remained unchanged and robust (**Fig 9**).

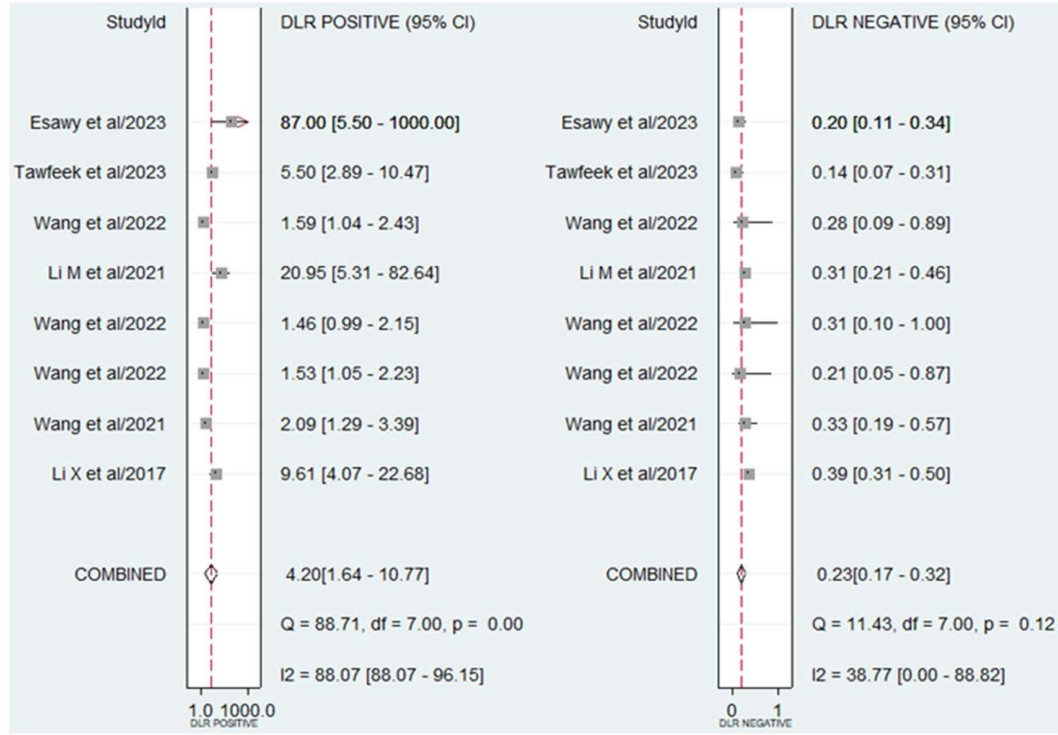

**Fig 4. Forest plots showing the PLR and NLR of lncRNAs in the diagnosis of AS.**

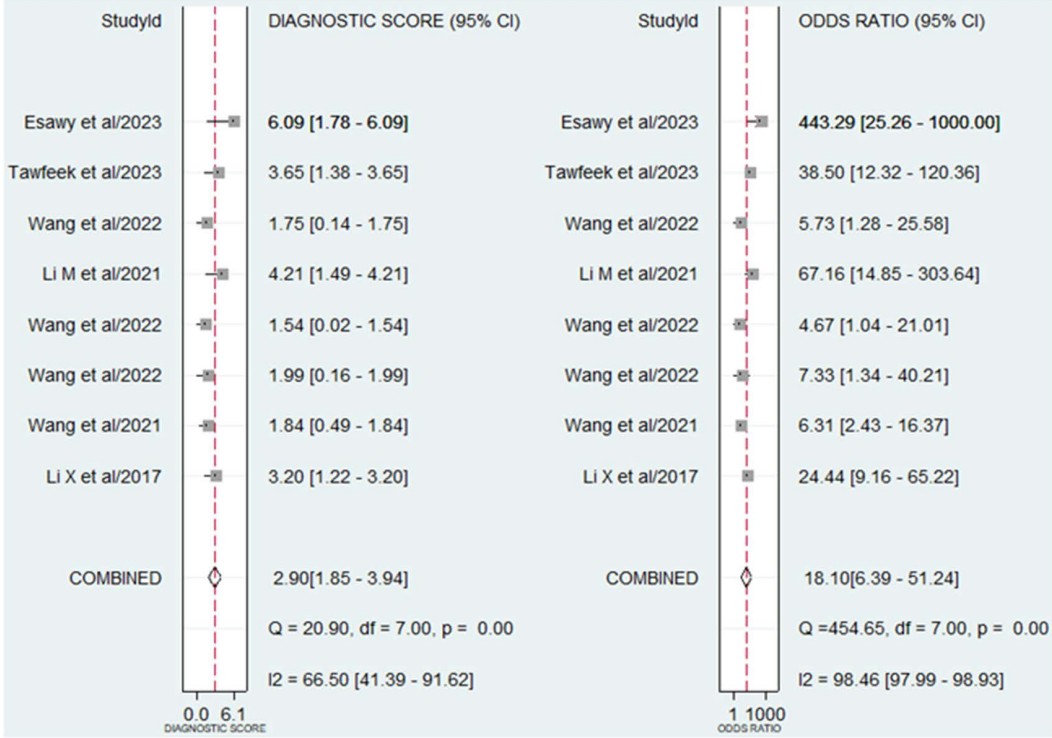

**Fig 5. Forest plots showing the DOR of lncRNAs in the diagnosis of AS.**

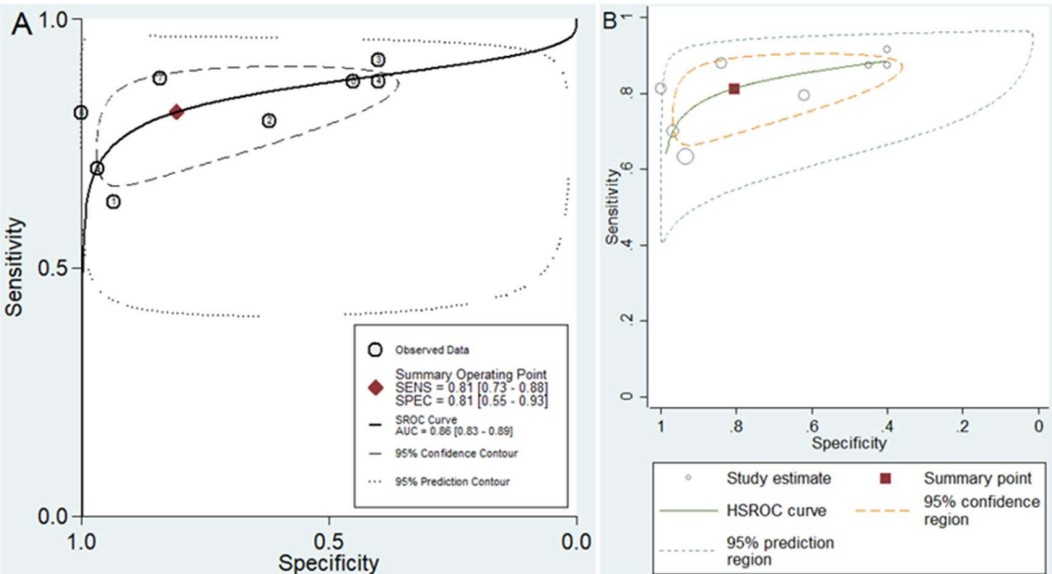

**Fig 6. SROC curve of lncRNAs for diagnosing AS, comprising (A) the SROC curve and (B) the HSROC model.**

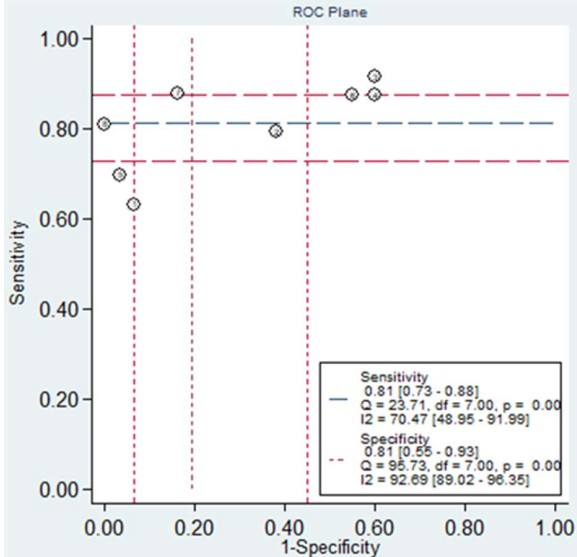

**Fig 7. ROC plane for the assessment of threshold effect.**

## Publication bias

The possible publication bias in the included studies was evaluated using Deeks' funnel plot asymmetry test. Overall, symmetry was observed in the funnel plots, and the slope coefficient was associated with a P value of 0.2, clearly demonstrating that no significant publication bias existed in this meta-analysis (**Fig 10**).

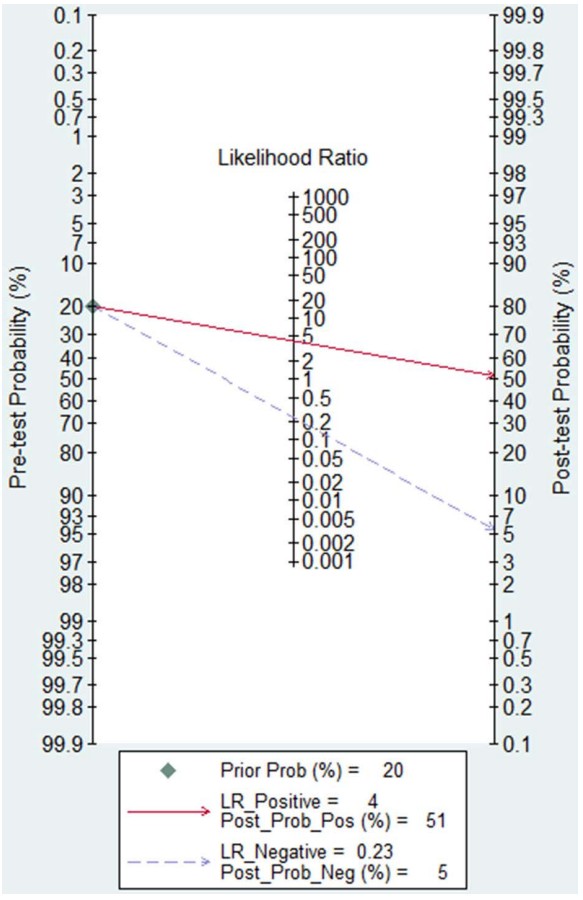

**Fig 8. Assessment of the clinical utility of lncRNAs in diagnosing AS using Fagan's nomogram.**

**Table 2. Subgroup analysis of the diagnostic potential of lncRNAs in AS.**

| Subgroup | No of studies | Sen (95% CI) | Spe (95% CI) | PLR (95% CI) | NLR (95% CI) | DOR (95% CI) | AUC (95% CI) |
|---|---|---|---|---|---|---|---|
| **Country** | | | | | | | |
| China | 6 | 0.80 (0.69−0.88) | 0.71 (0.42−0.89) | 2.7 (1.3−5.8) | 0.28 (0.21−0.38) | 10 (4−24) | 0.83 (0.79−0.86) |
| Egypt | 2 | − | − | − | − | − | − |
| **Specimen** | | | | | | | |
| PMBCs | 5 | 0.83 (0.74−0.89) | 0.63 (0.33−0.85) | 2.2 (1.1−4.5) | 0.27 (0.18−0.40) | 8 (3−22) | 0.84 (0.80−0.87) |
| Others | 3 | − | − | − | − | − | − |
| **Regulation mode** | | | | | | | |
| Upregulated | 7 | 0.83 (0.76−0.88) | 0.88 (0.47−0.94) | 3.8 (1.3−11.0) | 0.22 (0.16−0.30) | 18 (5−61) | 0.86 (0.82−0.88) |
| Downregulated | 1 | − | − | − | − | − | − |
| **References** | | | | | | | |
| β-actin | 6 | 0.82 (0.72−0.89) | 0.74 (0.40−0.92) | 3.1 (1.1−8.7) | 0.25 (0.16−0.38) | 13 (4−44) | 0.85 (0.81−0.88) |
| Others | 2 | − | − | − | − | − | − |

Note: PBMCs: peripheral blood mononuclear cells.

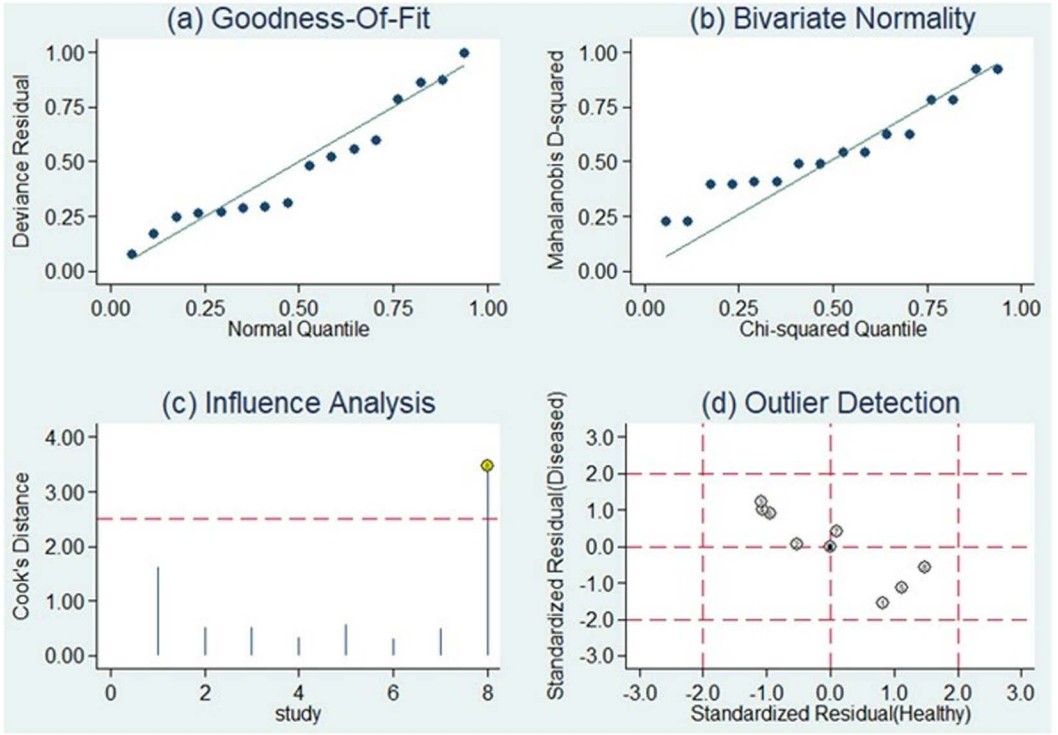

**Fig 9. The sensitivity analysis, aimed at evaluating the robustness of the meta-analysis results, involved a comprehensive assessment encompassing four key components: (a) goodness of fit, (b) bivariate normality, (c) influence analysis, and (d) outlier detection.**

## Discussion

In this meta-analysis, we comprehensively evaluated the diagnostic performance of lncRNAs in identifying individuals with AS. Our findings reveal promising diagnostic utility. The combined sensitivity and specificity of lncRNAs were estimated at 0.81 (95% CI, 0.73–0.88) and 0.81 (95% CI, 0.55–0.93), respectively, demonstrating a balanced ability to correctly identify both affected and unaffected individuals. The PLR of 4.2 (95% CI, 1.64–10.77), indicating that individuals with AS are 4.2 times more likely to test positive for lncRNAs than those without AS. Conversely, the NLR of 0.23 (95% CI, 0.17–0.32) suggests a 77% reduction in the odds of AS following a negative test result. The DOR, a comprehensive indicator of test performance, was 18.1 (95% CI, 6.39–51.24), highlighting the strong discriminatory capacity of lncRNAs. Additionally, the AUC of 0.86 (95% CI, 0.83–0.89) indicates excellent overall diagnostic accuracy. These findings highlight the potential of lncRNAs as non-invasive biomarkers for the diagnosis of AS that could complement existing diagnostic modalities, addressing the limitations of current methods such as imaging and serological markers.

The paramount importance of biomarkers lies in their ability to shape clinical decision-making processes. In our study, we employed a Fagan plot to assess the clinical significance of lncRNAs as diagnostic markers for AS. With the pre-test probability set at 20%, we observed that a positive test result significantly elevated the post-test probability to 51%, with a corresponding PLR of 4.2. This outcome emphasizes the effectiveness of lncRNAs in increasing the likelihood of diagnosing AS following a positive test outcome. Conversely, a negative test result, characterized by a NLR of 0.23, notably decreased the post-test probability to 5%. These findings underscore the potential of lncRNA testing to reliably exclude AS in instances where the test result is negative.

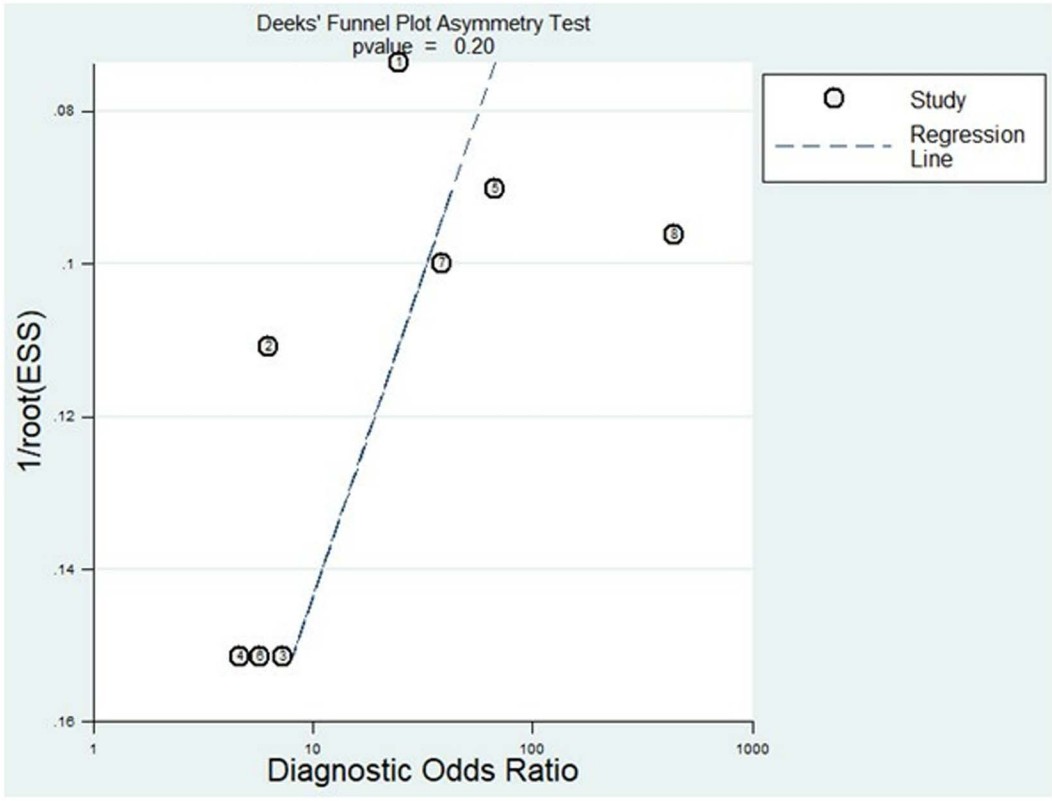

**Fig 10. The assessment of publication bias using Deek's funnel plot.**

Furthermore, our study underwent rigorous validation processes to ensure the reliability of our findings. We conducted both a goodness-of-fit analysis and bivariate normality analysis, which confirmed the robustness of our model. Sensitivity analysis was employed to detect potential outliers, yet no such outliers were identified, indicating the consistency and resilience of our aggregated results. To evaluate possible publication bias among the included studies, we utilized Deeks' funnel plot asymmetry test. Notably, the funnel plots demonstrated symmetry, and the associated slope coefficient yielded a P value of 0.2, signifying the absence of significant publication bias in our meta-analysis. These comprehensive analyses collectively reinforce the reliability and integrity of our study outcomes, underscoring the validity of our conclusions and the credibility of our analytical methodologies.

Our findings culminate in suggesting a promising role for lncRNAs as potential non-invasive biomarkers for diagnosing AS. The strength of our study lies in its comprehensive analysis of the diagnostic potential of lncRNAs for ankylosing spondylitis, the inclusion of multiple lncRNAs, utilizing robust statistical methods and a systematic approach to consolidate evidence from multiple studies. Additionally, the focus on non-invasive biomarkers provides valuable insights into improving early diagnosis and clinical management of AS. However, it is imperative to acknowledge the limitations inherent in our study. Firstly, the small sample sizes and limited number of studies may compromise the robustness of our conclusions. Secondly, variations in cutoff values, sample size, and internal reference control of lncRNAs across studies could introduce heterogeneity into our results. Thirdly, the inclusion of only certain countries may restrict the global applicability of lncRNAs' diagnostic performance for AS. Fourthly, the scarcity of similar lncRNAs for pooling results impedes the identification of specific single lncRNA or lncRNA panels as optimal diagnostic biomarkers for AS. Additionally, due to the limited number of studies, we refrained from conducting

meta-regression to explore the sources of heterogeneity. Hence, while our findings offer valuable insights, it is prudent to exercise caution in their interpretation.

In conclusion, the available evidence suggests that lncRNAs possess significant diagnostic utility in forecasting AS and can function as efficient non-invasive markers for the condition. However, the results should undergo further validation through well-designed longitudinal studies with larger sample sizes in the future to enhance their reliability and generalizability. Moreover, collaborative efforts among researchers globally could aid in overcoming the limitations and advancing our understanding of lncRNAs' diagnostic potential in AS. Incorporating lncRNA testing into clinical practice could facilitate early detection and personalized management of AS, potentially improving patient outcomes and reducing healthcare burdens associated with delayed diagnosis.

## Supporting information

**S1 Checklist. PRISMA 2020 Checklist.**
(DOCX)

**S1 Table. Search strategy used to retrieve eligible studies.**
(DOCX)

**S2 Table. A comprehensive list of all data extracted from the primary research sources.**
(DOCX)

**S3 Table. List of all studies assessed for eligibility criteria.**
(DOCX)

## Author contributions

**Conceptualization:** Ermiyas Alemayehu.

**Data curation:** Ermiyas Alemayehu, Sintayehu Ambachew, Daniel Asmelash, Melaku Ashagrie Belete.

**Formal analysis:** Ermiyas Alemayehu, Sintayehu Ambachew, Daniel Asmelash, Melaku Ashagrie Belete.

**Investigation:** Ermiyas Alemayehu, Sintayehu Ambachew, Daniel Asmelash, Melaku Ashagrie Belete.

**Methodology:** Ermiyas Alemayehu, Sintayehu Ambachew, Daniel Asmelash, Melaku Ashagrie Belete.

**Project administration:** Ermiyas Alemayehu, Sintayehu Ambachew, Daniel Asmelash, Melaku Ashagrie Belete.

**Resources:** Daniel Asmelash, Melaku Ashagrie Belete.

**Software:** Ermiyas Alemayehu, Sintayehu Ambachew, Daniel Asmelash, Melaku Ashagrie Belete.

**Supervision:** Ermiyas Alemayehu, Sintayehu Ambachew, Daniel Asmelash, Melaku Ashagrie Belete.

**Validation:** Ermiyas Alemayehu, Sintayehu Ambachew, Daniel Asmelash, Melaku Ashagrie Belete.

**Visualization:** Ermiyas Alemayehu, Sintayehu Ambachew, Daniel Asmelash, Melaku Ashagrie Belete.

**Writing – original draft:** Ermiyas Alemayehu, Melaku Ashagrie Belete.

**Writing – review & editing:** Ermiyas Alemayehu, Sintayehu Ambachew, Daniel Asmelash, Melaku Ashagrie Belete.

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
