## [Decision Letter · Decision Letter 0]

PONE-D-24-30235Diagnostic value of long noncoding RNAs as biomarkers for ankylosing spondylitis: A systematic review and meta-analysisPLOS ONE

Dear Dr. Alemayehu,

Thank you for submitting your manuscript to PLOS ONE. After careful consideration, we feel that it has merit but does not fully meet PLOS ONE’s publication criteria as it currently stands. Therefore, we invite you to submit a revised version of the manuscript that addresses the points raised during the review process.

We look forward to receiving your revised manuscript.

Kind regards,

Shaghayegh Khanmohammadi

Academic Editor

PLOS ONE

Journal Requirements:

https://www.frontiersin.org/journals/immunology/articles/10.3389/fimmu.2023.1131355/full

https://dr.ntu.edu.sg/bitstream/10356/161475/2/fimmu-13-790924.pdf

In your revision ensure you cite all your sources (including your own works), and quote or rephrase any duplicated text outside the methods section. Further consideration is dependent on these concerns being addressed.

4. As required by our policy on Data Availability, please ensure your manuscript or supplementary information includes the following: 

5. Please include captions for your Supporting Information files at the end of your manuscript, and update any in-text citations to match accordingly. Please see our Supporting Information guidelines for more information: http://journals.plos.org/plosone/s/supporting-information

Reviewers' comments:

Reviewer's Responses to Questions

**Comments to the Author**

1. Is the manuscript technically sound, and do the data support the conclusions?

Reviewer #1: Yes

Reviewer #2: Yes

2. Has the statistical analysis been performed appropriately and rigorously? 

Reviewer #1: Yes

Reviewer #2: Yes

3. Have the authors made all data underlying the findings in their manuscript fully available?

Reviewer #1: Yes

Reviewer #2: No

4. Is the manuscript presented in an intelligible fashion and written in standard English?

Reviewer #1: No

Reviewer #2: Yes

5. Review Comments to the Author

Reviewer #1: It is interestin title, I have seen only two coutries if possible please include and investigate additional countries/ write only china and exclude Egypt ? it is better that writing in the whole country or only china and please reanalyze again

Reviewer #2: Alemayehu et al. have performed a systematic review and meta-analysis of lncRNAs as biomarkers for AS. By inclusion of 11 studies, they showed a high diagnostic ability of them. The manuscript is well-written, however, I have the following concerns:

- Abstract: There is no need to mention the PROSPERO code in the introduction section of the abstract.

- Introduction: There is unnecessary information about AS in the introduction section. Instead, authors could focus on the details and controversial findings regarding lncRNAs in AS.

- The search was performed more than a year ago. It is highly suggested that the authors update their search for recent studies.

- The use of RevMan for QA using the QUADAS-2 tool is only mentioned in the results section. Details should be added to the methods section as well.

- The first paragraph of the discussion should focus on all the main findings of the manuscript.

- The strengths of the manuscript should be also mentioned in the discussion section.

- A paragraph summarizing the clinical take-home message of this manuscript should be added to the discussion.

6. PLOS authors have the option to publish the peer review history of their article (what does this mean? ). If published, this will include your full peer review and any attached files.

**Do you want your identity to be public for this peer review?** For information about this choice, including consent withdrawal, please see our Privacy Policy .

Reviewer #1: No

Reviewer #2: No

---

## [Author Response · Author response to Decision Letter 1]

10 Apr 2025

Authors’ Response to Reviewer comments:

Thank you for your time and for critically reviewing our manuscript which helps us to render better clarity to the paper and make it scientifically plausible. Sorry for the delay of our response. It is because of a blackout of internet connection in our region due to ongoing war.

All the raised questions have been addressed; and the manuscript is modified accordingly. Moreover, all the requested corrections are addressed in both the revised manuscript and the response letter. Changes are shown with track changes in the file labeled ‘Revised Manuscript with Track Changes’. The point-by-point response is given below.

Journal editorial requirements:

Author’s Response:

Thanks for the significant editorial comment. The revised manuscript is now updated and meets PLOS ONE's style requirements, including those for file naming.

https://www.frontiersin.org/journals/immunology/articles/10.3389/fimmu.2023.1131355/full

https://dr.ntu.edu.sg/bitstream/10356/161475/2/fimmu-13-790924.pdf

In your revision ensure you cite all your sources (including your own works), and quote or rephrase any duplicated text outside the methods section. Further consideration is dependent on these concerns being addressed.

Author’s Response:

Thank you for the feedback and we have managed the overlapping texts from previous publications.

Author’s Response:

We have proofread the manuscript for any language errors and typos.

4. As required by our policy on Data Availability, please ensure your manuscript or supplementary information includes the following:

Name of data extractors and date of data extraction

Confirmation that the study was eligible to be included in the review.

All data extracted from each study for the reported systematic review and/or meta-analysis that would be needed to replicate your analyses.

If data or supporting information were obtained from another source (e.g. correspondence with the author of the original research article), please provide the source of data and dates on which the data/information were obtained by your research group.

If applicable for your analysis, a table showing the completed risk of bias and quality/certainty assessments for each study or outcome.

An explanation of how missing data were handled.

This information can be included in the main text, supplementary information, or relevant data repository.

Author’s Response:

Thank you and we have prepared and added the supplementary file in accordance with the new PLOS ONE policy on Data Availability.

5. Please include captions for your Supporting Information files at the end of your manuscript, and update any in-text citations to match accordingly. Please see our Supporting Information guidelines for more information: http://journals.plos.org/plosone/s/supporting-information

Author’s Response:

We have provided proper citations of all supporting information files in the manuscript.

Reviewer #1 comments:

1. It is interesting title; I have seen only two countries, if possible, please include and investigate additional countries/ write only China and exclude Egypt? it is better that writing in the whole country or only China and please reanalyze again

Author’s Response:

We appreciate the suggestion and agree that expanding the geographical scope of the study would strengthen its applicability. However, due to limited availability of relevant studies from other countries, we included studies only from China and Egypt. It is evident that majority of included studies span from China, but we still included pertinent data regarding performances of two lncRNAs (TUG1 and H19) from Egypt and we incorporated them in our analysis. Our main purpose was to provide a preliminary and comprehensive data regarding the diagnostic value of lncRNAs as biomarkers for ankylosing spondylitis, thus we included all eligible studies in our analysis.

Reviewer #2 comments:

1. Abstract: There is no need to mention the PROSPERO code in the introduction section of the abstract.

Author’s Response:

We appreciate the valuable feedback and in response we remove the PROSPERO code from the abstract section in our revision (Lines 31-32).

2. Introduction: There is unnecessary information about AS in the introduction section. Instead, authors could focus on the details and controversial findings regarding lncRNAs in AS.

Author’s Response:

Thank you and we have revised the introduction by reducing background information on AS and emphasizing the controversial findings surrounding lncRNAs as biomarkers in AS. In our revision, we deleted general descriptions about AS that are not directly relevant to the study, and highlighted evidence supporting the diagnostic value of lncRNAs, emphasizing gaps in current understanding (Lines 59-90).

3. The search was performed more than a year ago. It is highly suggested that the authors update their search for recent studies.

Author’s Response:

Thank you for the valuable comment. We have performed an updated search (on 15th March 2025) from all databases but we were not able to get new eligible studies. We have revised Supplementary Table 2 and the PRISMA flow chart based on the most recent search results (Line 122, 200-205).

4. The use of RevMan for QA using the QUADAS-2 tool is only mentioned in the results section. Details should be added to the methods section as well.

Author’s Response:

We are grateful for the suggestion and we have expanded the methods section to include a detailed description of the Quality Assessment of Diagnostic Accuracy Studies-2 (QUADAS-2) tool and the use of RevMan software for quality assessment. This ensures consistency between the methods and results sections (Lines 158-164).

5. The first paragraph of the discussion should focus on all the main findings of the manuscript.

Author’s Response:

Thank you and we have deleted the first two paragraphs of the discussion section incorporating their summarized points into the introduction section. Additionally, the new first paragraph of the discussion has been revised to summarize the primary findings, including the diagnostic accuracy of lncRNAs and their potential as non-invasive biomarkers for AS (Lines 325-351).

6. The strengths of the manuscript should be also mentioned in the discussion section.

Author’s Response:

Thank you for the feedback. A dedicated paragraph has been added to the discussion, emphasizing the strengths of the study, such as the systematic approach, the inclusion of multiple lncRNAs, and the use of robust statistical methods to assess diagnostic accuracy (Lines 392-396).

7. A paragraph summarizing the clinical take-home message of this manuscript should be added to the discussion.

Author’s Response:

Thank you and we have added a concluding paragraph at the end of the discussion section that summarizes the clinical implications, highlighting the potential of lncRNAs as reliable, non-invasive diagnostic tools for AS and their role in improving early detection (Lines 412-414).

Responses to Reviewer 1 or 2 (from the attached word document):

1. From Lines 56-65 it is better to write about how Long noncoding RNAs (lncRNAs) have emerged as promising biomarkers for various diseases, including ankylosing spondylitis (AS), a chronic inflammatory condition primarily affecting the spine and sacroiliac joints A systematic review and meta-analysis could address your interest in the diagnostic value of lncRNAs for AS. Here's a broad outline of what such a study might entail so please improve the these points

Author’s Response:

Thank you for your insightful suggestion. We have revised the manuscript to highlight the promising potential of lncRNAs as non-invasive diagnostic tools, which is now addressed at the end of the third paragraph (Lines 88-90).

2. The introduction needs revision focused what you want to write Diagnostic value of long noncoding RNAs as biomarkers for ankylosing 2 spondylitis

Author’s Response:

Thank you and we have revised the Introduction section thoroughly deleting basic information and focusing on the diagnostic efficacy of lncRNAs for AS (Lines 59-90).

3. The objective is better to write like To evaluate the diagnostic utility of long noncoding RNAs (lncRNAs) as biomarkers for ankylosing spondylitis (AS) by systematically reviewing and analyzing existing literature (optional not mandatory).

Author’s Response:

Comment accepted and we revised the objective accordingly (Lines 101-102).

4. Methods: Line 111 Study Design or registration?

Author’s Response:

Thank you for the feedback, and we revised the subtopic as “Study protocol registration” (Line 113).

5. Methods: Databases: PubMed, Embase, Cochrane Library, Web of Science.

Author’s Response:

We appreciate the valuable feedback regarding the scope of our systematic search. In our study, we included searches through PubMed, Embase, Scopus, and Hinari, which we believed to be comprehensive sources for identifying relevant studies. The omission of Web of Science was not intentional, but rather due to the lack of access (institutional subscription) to Web of Science. To compensate for it, we have included other sources (direct Google search, Google Scholar, ResearchGate) and traced bibliographies of included articles to include pertinent studies inadvertently overlooked in the initial search. However, we did not use Cochrane Library as it primarily focuses on nursing related studies.

6. Keywords: "long noncoding RNAs", "lncRNAs", "ankylosing spondylitis", "AS", "biomarkers", "diagnostic value".

Author’s Response:

Thank you and we revised the key words based upon the comment. We have re-run our search based on the new key word combinations however the search result remained the same (Line 125).

7. Methods: Study period?

Author’s Response:

We have revised the study period as our revised search included eligible studies published until March 15, 2025. Additionally, we also incorporate the study period in the abstract section for further clarity (Line 122).

8. Summary Table: Overview of included studies, lncRNAs investigated, and their results showed that China and EGYPT. Is only those country? please see these review reference A review of long non-coding RNAs in ankylosing spondylitis: pathogenesis, clinical assessment, and therapeutic targets

Author’s Response:

We appreciate the feedback and unfortunately our search identified eligible studies only from two countries (China and Egypt) despite our in-depth global search. Although similar studies were available from other parts of the world, these studies were not included in our analysis since they were not eligible or fulfill the eligibility criteria.

9. Is Subgroup analysis of the diagnostic potential of lncRNAs in AS is possible by only two Groups?

Author’s Response:

We have performed subgroup analysis based on various specific groups such as country, type of specimen, regulation mode, and experimental references. We chose these segments for subgroup analysis based on the information provided by the eligible studies.

---

## [Decision Letter · Decision Letter 1]

Diagnostic value of long noncoding RNAs as biomarkers for ankylosing spondylitis: A systematic review and meta-analysis

PONE-D-24-30235R1

Dear Dr. Alemayehu,

We’re pleased to inform you that your manuscript has been judged scientifically suitable for publication and will be formally accepted for publication once it meets all outstanding technical requirements.

Kind regards,

Shaghayegh Khanmohammadi

Academic Editor

PLOS ONE

Additional Editor Comments (optional):

Reviewers' comments:

Reviewer's Responses to Questions

**Comments to the Author**

1. If the authors have adequately addressed your comments raised in a previous round of review and you feel that this manuscript is now acceptable for publication, you may indicate that here to bypass the “Comments to the Author” section, enter your conflict of interest statement in the “Confidential to Editor” section, and submit your "Accept" recommendation.

Reviewer #2: All comments have been addressed

2. Is the manuscript technically sound, and do the data support the conclusions?

Reviewer #2: (No Response)

3. Has the statistical analysis been performed appropriately and rigorously? 

Reviewer #2: (No Response)

4. Have the authors made all data underlying the findings in their manuscript fully available?

Reviewer #2: (No Response)

5. Is the manuscript presented in an intelligible fashion and written in standard English?

Reviewer #2: (No Response)

6. Review Comments to the Author

Reviewer #2: (No Response)

7. PLOS authors have the option to publish the peer review history of their article (what does this mean? ). If published, this will include your full peer review and any attached files.

**Do you want your identity to be public for this peer review?** For information about this choice, including consent withdrawal, please see our Privacy Policy .

Reviewer #2: No

---

## [Editor Report · Acceptance letter]

PONE-D-24-30235R1

PLOS ONE

Dear Dr. Alemayehu,

I'm pleased to inform you that your manuscript has been deemed suitable for publication in PLOS ONE. Congratulations! Your manuscript is now being handed over to our production team.

Kind regards,

on behalf of

Dr. Shaghayegh Khanmohammadi

Academic Editor

PLOS ONE